# New Approaches for *Escherichia coli* Genotyping

**DOI:** 10.3390/pathogens9020073

**Published:** 2020-01-21

**Authors:** Roman Kotłowski, Katarzyna Grecka, Barbara Kot, Piotr Szweda

**Affiliations:** 1Department of Microbiology and Molecular Biotechnology, Faculty of Chemistry, Gdańsk University of Technology, Str. G. Narutowicza 11/12, 80-233 Gdańsk, Poland; 2Department of Pharmaceutical Technology and Biochemistry, Faculty of Chemistry, Gdańsk University of Technology, Str. G. Narutowicza 11/12, 80-233 Gdańsk, Poland; kagrecka@gmail.com; 3Siedlce University of Natural Sciences and Humanities, Faculty of Exact and Natural Sciences, Institute of Biological Sciences, 14 Bolesława Prusa Str., 08-110 Siedlce, Poland; barbara.kot@uph.edu.pl

**Keywords:** *Escherichia coli*, BOX-PCR, *fliC*, polymerase chain reaction-restriction fragment length polymorphism (PCR-RFLP), genotyping

## Abstract

Easy-to-perform, fast, and inexpensive methods of differentiation of *Escherichia coli* strains beyond the species level are highly required. Herein two new, original tools for genotyping of *E. coli* isolates are proposed. The first of the developed method, a PCR-RFLP (polymerase chain reaction-restriction fragment length polymorphism) test uses a highly variable *fliC* gene, encoding the H antigen as a molecular target. The designing of universal pair of primers and selection of the optimal restriction enzyme *Rsa*I was preceded by in silico comparative analysis of the sequences of the genes coding for 53 different serotypes of H-antigen (*E. coli* flagellin). The target fragments of *E. coli* genomes for MLST method were selected on the basis of bioinformatics analysis of complete sequences of 16 genomes of *E. coli*. Initially, seven molecular targets were proposed (seven pairs of primers) and five of them were found useful for effective genotyping of *E. coli* strains. Both developed methods revealed high differentiation power, and a high genetic diversity of the strains tested was observed. Within the group of 71 strains tested, 29 and 47 clusters were revealed with *fliC* RFLP-PCR and MLST methods, respectively. Differentiation of the strains with the reference BOX-PCR method revealed 31 different genotypes. The in silico analysis revealed that the discriminatory power of the new MLST method is comparable to the Pasteur and Achtman schemes and is higher than the discriminatory power of the method developed by Clermont. From the epidemiology point of view, the outcomes of our investigation revealed that in most cases, the patients were infected with unique strains, probably from environmental sources. However, some strains isolated from different patients of the wards of pediatrics, internal medicine, and neurology were classified to the same genotype when the results of all three methods were taken into account. It could suggest that they were transferred between the patients.

## 1. Introduction

Typing of microbial pathogens or identifying bacteria at the strain level is important for epidemiological studies but also the diagnosis and treatment of bacterial infections. This is a particularly difficult but important challenge in the case of those types of bacteria that are characterized by high genetic variability (which leads to important differences in virulence potential) but are also very common in the environment, such as *Escherichia coli* and other *Enterobacteriaceae*. 

*E. coli* is a ubiquitous member of the normal intestinal bacterial microflora in humans, other warm-blooded animals, and reptiles. The natural environment of these bacteria is the mucous layer of the cecum and colon [1,2]. However, in the course of evolution, some *E. coli* clones acquired genetic elements that enabled them to colonize different ecological niches (water, soil, different biotic and abiotic surfaces) but also transformed some of them into pathogenic strains [3]. Depending on the location of the infection, pathogenic *E. coli* strains can be divided into two basic groups: intestinal pathogenic *E. coli* (IPEC) and extraintestinal pathogenic *E. coli* (ExPEC) [1,2,4]. Within the group of IPEC, the most important are enteroaggregative *E. coli* (EAEC), enterohaemorrhagic *E. coli* (EHEC), enteroinvasive *E. coli* (EIEC), enteropathogenic *E. coli* (EPEC), enterotoxigenic *E. coli* (ETEC), diffusely adherent *E. coli* (DAEC), and adherent invasive *E. coli* (AIEC). Uropathogenic *E. coli* (UPEC), meningitis-associated *E. coli* (MNEC), septicemia-associated *E. coli* (SEPEC), and avian pathogenic *E. coli* (APEC) are the most common ExPEC pathotypes [1,2,4,5]. The large diversity of phenotypic properties of these bacteria, which manifests as differences in their pathogenicity or as the ability to grow in different environmental conditions, is of course due to differences in the content of their genomes. Much effort has been put into research on the immunological, phenotypic, or genetic variability of this group of bacteria.

In 1984, Ochman and Selander established a reference group of 72 strains of *E. coli* (ECOR; Escherichia coli reference collection). The selected isolates were recovered from human and 16 other mammalian hosts from various geographical regions mainly from the USA and Sweden. By means of the multilocus enzyme electrophoresis technique, the collection of strains was divided into four major phylogenetic subgroups: A, B1, B2, and D. Up to date, this division is the basis for classification of *E. coli* [6]. Based on many years of research, some regularities have been observed regarding the relationship between belonging to a specific subgroup and the ability to cause the disease. Commensal bacteria, a component of the natural microflora, most often belong to subgroup B1 or A. Bacteria causing parenteral infections, mainly in the urinary tract, are most often classified in subgroup B2, rarely D. The greatest diversity was observed among bacteria causing mild or chronic diarrhea. In this case, the isolates may belong to all four phylogenetic groups [7]. The division proposed by Ochman and Selander, however useful, does not solve the problem of phylogenetic analysis of *E. coli*. Within each of the groups of these bacteria, a strong genetic diversity of isolates is observed. However, to date, many different techniques have been proposed for the analysis of genetic versatility of *E. coli* isolates; it is still necessary to develop new fast, easy, reproducible, and inexpensive methods of genotyping of this group of bacteria that can be used in most, even not well-equipped, laboratories.

Herein we propose two new, original tools that can be useful for genotyping of *E. coli* isolates. The first of the developed method, a PCR-RFLP (polymerase chain reaction-restriction fragment length polymorphism) test uses a highly variable *fli*C gene [8,9,10], encoding the H antigen as a molecular target. The second approach is based on the PCR amplification of seven selected genes that characterize with high sequence and size variability within the sequenced genomes of *E. coli* [11]. The outcome of our study confirmed the high differentiation potential of both the proposed methods.

## 2. Results

### 2.1. BOX-PCR

Initially, the genetic variability of *E. coli* strains tested were investigated with the BOX-PCR method described previously [12,13,14,15]. The results of this part of the research were used as a reference (base) for the assessment of the discriminatory power of the two new methods of genotyping of *E. coli*. Amplification with BOX primer [14,16] was successful for all strains tested, with at least two products—DNA fragments (Figure 1). The results of the cluster analysis are presented in Figure 2. Thirty-one different genotypes were identified, among them, 21 genotypes were represented by only one strain. The largest cluster contained 19 strains, almost all isolated in 2008. Eleven, three, three, and two of them were recovered from the patients of the pediatric, obstetric, neurology, and internal units, respectively.

The second-largest genotype contained nine strains, five isolated in 2008 (four from the pediatric unit and one from dialysis unit); the following four strains were isolated in 2007 (two from patients of the internal ward and the two following strains were from internal and neurology units, respectively). Two genotypes consisted of four isolates. Interestingly, only in the case of one of them, two strains were isolated from the same ward (internal) in the same year (2008); other isolates were collected in different places (units of the hospital) or time. Similarly, no evident correlation between place/time of isolation and classification of the strains to the genotypes was observed either for two genotypes that consisted of three isolates (only two strains of one cluster were collected in a pediatric unit in 2008) nor for three genotypes that consisted of two strains. The last two isolates classified to the fourth cluster consisted of two strains that were recovered from the pediatric unit in 2008.

### 2.2. FliC PCR-RFLP

Sequences from all H-antigens (flagellin)-encoding genes, *fliC* and their homologs (in total 53 sequences—Appendix A) [17,18] were used for comparative bioinformatic analysis and designing of universal primers that could be used for amplification of variable fragments of this molecular target (the comparison of sequences of reference strains and location of universal primers is presented in Appendix A). For all strains tested, the PCR amplification with the designed pair of primers was successful. Digestion of the obtained PCR products with the *Rsa*I restriction enzyme revealed 29 different genotypes. Clear and easily interpretable results are the main advantage of the proposed PCR-RFLP method (Figure 3). The largest cluster consisted of 19 strains, and among them, 15 strains were isolated in 2008 from patients hospitalized in differed wards of hospital (ten strains from children, three from the patients hospitalized in the neurological ward and one strain from a dialysis patient and one from an internal medicine patient). Four strains representing this genotype (two from internal medicine patients and one from a patient hospitalized in the neurological ward and one from pregnant women) were isolated in 2007. The second-largest genotype consisted of seven strains, among them, six were isolated in 2008. Three strains belonging to this genotype were isolated from neurological patients, the next three from internal medicine patients, and one strain from a patient hospitalized in the orthopedics ward. The last member of this group was collected in 2007 from the patient of the neurology unit. Five strains (three isolated from children in 2008 and two isolated from internal medicine patients in 2007) formed another cluster. The cluster analysis also revealed three genotypes consisting of four strains. One of these clusters was very diverse from the point of view of the origin of the strains—each of them was isolated from one patient hospitalized in different hospital wards. A second genotype consisting of four strains isolated in 2008 included three strains from pregnant women and one from a child. The third genotype (counting four strains) included two strains isolated from children (2008) and two isolated from internal medicine patients (2007). The three strains that formed the next cluster also differed in isolation source. Interesting results were obtained in the case of the last three genotypes that consisted of two strains each. In all cases, the strains classified to one genotype were collected from the same source. The remaining 19 strains were different from other strains and comprised separate genotypes (Figure 4). High differentiation power of the proposed *fliC* RFLP-PCR method has also been confirmed by the bioinformatic simulation conducted for sequences of all 53 H-antigen (flagellin) encoding genes. As a result of in silico RFLP-PCR differentiation of these sequences, 51 different genotypes were identified (Appendix A). The proposed approach with the application of *Rsa*I restrictase did not differentiate only two pairs of H-antigen genotypes, namely H4 –H25 and H1–H12. However, within both pairs, some differences of the nucleotide sequences of the *fliC* gene have been identified, and enzymes other than *Rsa*I can be proposed for differentiation of H4 –H25 and H1–H12 genotypes (Appendix A).

### 2.3. Multilocus Sequence Typing

On the basis of bioinformatic analysis of 16 *E. coli* genomes deposited in GenBank, a set of seven different pairs of primers was proposed for differentiation of *E. coli* strains tested [11]. PCR amplification was performed separately for each pair of designed primers. In all cases, positive results of amplification for all tested strains were obtained. Some important differences in discriminatory power between these pairs of primers have been observed (Figure 5, Figure 6, Figure 7, Figure 8 and Figure 9). In the case of pairs of primers Ec3–Ec4 and Ec7–Ec8, the discriminatory power was very low (data not shown). Moreover, we observed some problems with the reproducibility of the results of the PCR reaction performed with these pairs of oligonucleotides. A medium level of discriminatory power was observed in the case of PCR reactions carried out with pairs of primers Ec1–Ec2 and Ec9–Ec10 (Figure 5 and Figure 7). Three remaining PCR reactions with pairs of primers Ec5–Ec6, Ec11–Ec12, and especially Es13–Ec14 showed high discriminatory power. The final differentiation of the tested strains was performed on the basis of combination of results of five PCR reactions that were conducted with five pairs of primers (multilocus sequence typing): Ec1–Ec2, Ec5–Ec6, Ec9–Ec10, Ec11–Ec12, Ec13–Ec14. The results of the cluster analysis are presented in Figure 10. The typing revealed nine genotypes including from two to seven strains each and 38 genotypes that consisted of only one strain. The largest cluster included strains from five different wards, namely pediatrics (two strains, 2008) internal (two strains, 2007), dialysis, obstetrics, and neurology (one strain in the case of each of them). No important relationship between cluster classification and place/time of isolation was observed in the case of the second-largest genotype consisting of five strains. This cluster included strains from patients of pediatrics (two strains, 2008), internal (two strains, 2007), and dialysis (one strain, 2008) wards. A similar situation was found in two out of five genotypes with three isolates each. In both cases, all strains were isolated in different units or at different times. On the other hand, only the strains from the pediatric unit, isolated in 2008, were classified into two following genotypes consisted of three strains. Two isolates from neurology and one from the internal unit were classified into the last genotype that covered three strains. Two strains collected from children and one from the patients of each neurology and internal wards (all isolated in 2008) were classified as the only cluster counting four strains. The only cluster counting two members covered strains from the patients of the neurology unit, isolated in 2008. Identification of 38 genotypes of only one strain confirmed the high discriminatory power of the proposed-herein MLTS method. 

## 3. Discussion

The clones of *E. coli* that acquired genes coding for virulence factors belong to the most important human and animal pathogens. Moreover, these bacteria are naturally present in the intestinal tracts of warm-blooded animals. Since *E. coli* is released into the environment through deposition of fecal material, this bacterium is widely used as an indicator of fecal contamination of water (in natural water reservoirs, e.g., lakes or rivers, but also tap water that is intended for human consumption) and food (it is important marker of hygiene in food processing plants) [19,20]. Thus fast, easy-to-perform, and inexpensive methods of detection but also differentiation of *E. coli* strains beyond the species level, are highly required. In this study, we have proposed two new methods that can be used for phylogenetic analysis of *E. coli* strains isolated from different sources, including human/animal infections, food, or environment samples (e.g., soil, water). Both these methods are based on PCR reaction. Thus, they are not very expensive or time-consuming and are easy in performance; no specialized equipment is required (only thermocycler and agarose gel electrophoresis). Another important advantage of proposed methods is high discriminatory power, which has been confirmed in the presented research. Both methods exhibit comparable (RFLP-PCR) or higher (MLST) potential for differentiation of the strains tested in comparison to the BOX-PCR method, which is widely used for genotyping of different pathogenic bacteria, including *E. coli* [15,21,22]. The differentiation of the strains tested with the BOX-PCR method revealed 31 different genotypes, while 29 and 47 clusters were revealed with the *fliC* RFLP-PCR and MLST methods, respectively. The discriminatory power of MLST is particularly high and in our opinion, this method deserves special attention. The target fragments of *E. coli* genomes were selected on the basis of bioinformatics analysis/comparison of complete sequences of 16 genomes of *E. coli*. Initially, seven molecular targets were proposed (seven pairs of primers) [11] that were verified in this study. In the case of two of them, some problems with the reproducibility of PCR amplification were observed. Moreover, the discriminatory power of these two targets was not very high. Amplification of other targets (five) was not problematic and discriminatory power was on a good level for each of them, particularly for target amplified with a set of primers Ec-13 and Ec-14. In all cases, the size of obtained PCR products was in the ranges predicted in in-silico simulation. The low discriminatory power of the above-mentioned two targets is also in agreement with the results of bioinformatics analysis [11]. On the basis of polymorphism of target two, amplified with primers Ec-3 and Ec-4, the group of 16 reference genomes was divided into five clusters; however, one of them covered ten strains and the size of amplified fragments were in the range between 114 and 195 bps. Only three genotypes were revealed in the case of the fragment of the genome that was amplified with Ec-7 and Ec-8 and the size of amplified fragments was in the range between 300 and 392 bps. On the other hand, in silico simulation with Ec-13 and Ec-14 primers revealed nine different genotypes and the largest of them consisted of only three strains; the size of amplified fragments ranged from 152 to 1132 bps—in vitro tests presented herein confirmed the high discriminatory potential of this fragment of *E. coli* genome. The final cluster analysis was performed by combining the results of the PCR reactions for five targets (the two “problematic” targets were omitted). As a result, a multilocus sequence typing (MLST) method was developed. Of course, this method can be improved, for example, by parallel amplification of selected DNA fragments in one reaction (multiplex PCR). However, based on our previous experience, we know that the simultaneous amplification of several DNA fragments in one tube can be troublesome. Usually a process of optimizing the conditions of amplification reaction is required, and some problems with the reproducibility of the reaction conducted in different laboratories or using different materials (e.g., DNA polymerases, buffers) can appear. In the proposed method, only five different targets are amplified. In our opinion, the divergence of even relatively large collections of strains (counting tens of strains) can be quickly analyzed with separate amplification of each of the targets, as was performed in this study.

The other of the developed methods, *fliC* RFLP-PCR, also can be recommended for genotyping and phylogenetic analysis of *E. coli* strains isolated from different sources. The designing of primers and selection of the optimal restriction enzyme *Rsa*I was preceded by in silico comparative analysis of 53 sequences of the whole genes *fliC* coding for the H antigen of *E. coli* [17,18]. The results of the in silico investigation, including designing of primers as well as simulation of cluster analysis of the products of digestion of PCR amplified fragments of all reference sequences of the *fliC* gene are presented in the Appendix A (Appendix A). To date, some authors proposed polymorphic regions of the *flicC* gene as a molecular target for the identification of specific groups of *E. coli*, e.g., *E. coli* O157:H7 and O157:NM [23]. Some interesting proposition of differentiation of clinical isolates of *E. coli* on the basis of differences in the sequences of this gene has been also proposed, e.g., Banjo and coworkers developed a complete and practical PCR-based H-typing system consisting of ten multiplex PCR kits with 51 single PCR primer pairs [18]. The most important advantage of the developed tests in this research method is the fact that the designed primers are universal for all strains of *E. coli*. Thus, it can be applied for both pathogenic and nonpathogenic *E. coli* isolates recovered from different sources. Moreover, the number of GC pairs (twelve) in the DNA sequences of both the primer sequences for the PCR method allows the use of highly specific hybridization of primers with the template DNA, even at a temperature of 69 °C. With such a high annealing temperature in the PCR reaction, it is possible to obtain only one specific fragment of DNA for further restriction analysis. The selection of the *Rsa*I enzyme [24] was also based on in silico trials and was crucial for the development of the method of high discriminatory power—comparable to BOX-PCR. 

The development of easy-to-perform and reliable methods of differentiation of the *fliC* gene is of special interest for clinical practice. Identification of pathogenic strains of *E. coli* is carried out by O:H serotyping based on the presence of the flagellar H and polysaccharide O antigens, and a specific antigen–antibody reaction. To date, 186 somatic O antigens and 53 flagellar H antigens are known [17,18,25]. This method, valued for its precision, reliability, and repeatability, is the primary technique used by diagnostic laboratories in the serotyping of clinical isolates of *E. coli*. However, it is not free from disadvantages. For example, all 56 specific sera are needed to detect a particular type of antigen, which generates a considerable cost. Another problem is the long time needed to perform the serotyping test and to obtain sera from animal blood. Obtaining antibodies requires prior vaccination (immunization) of the animals, as well as subsequent isolation from the blood and proper treatment. All these steps expose animals to stress and may even lead to their death. The herein proposed RFLP-PCR based method could be an interesting alternative for serotyping and identification of pathogenic *E. coli* isolates. According to the results of a in silico simulation with this method, 51 different genotypes (similarity 99% as a borderline) were identified within the group of 53 reference strains (Appendix A). Moreover, the two pairs of genotypes (H4–H25 and H1–H12) that were not differentiated with this method can be easily distinguished by the digestion of the PCR amplified fragment of the *fliC* gene with several, other than *Rsa*I, restriction enzymes (appropriate enzymes for both pairs of genotypes are proposed in the Appendix A).

All three methods revealed the high genetic diversity of the strains tested. Most of the identified genotypes, namely 19, 21, and 38 for the *fliC* RFLP-PCR, BOX-PCR and MLST methods, respectively, consisted of only one strain. The largest cluster, counting 19 strains, was identified in the case of BOX-PCR or *fliC* RFLP-PCR fingerprinting. However, only eleven strains belonged to both these clusters, namely 4, 8, 10, 18, 21, 34, 37, 50, 51, 58 and 61. With the MLST method, these strains were classified into five different clusters counting four (8, 10, 18, 37), three (4, 58, 61), two (21, 34) and one (two genotypes—isolates 51 and 50) strains, respectively. According to the MLST method, the similarity between these genotypes was very high—above 96%. The following eight strains, 13, 15, 32, 38, 40, 45, 59, and 63, from the largest cluster of *fliC* RFLP-PCR were classified to the second largest genotype of BOX-PCR, and seven of them, 13, 15, 32, 38, 40, 45, and 63 were found as the largest genotype of the MLST method. Thus these seven strains were classified into one genotype in all three methods. The isolate assigned with number 59 was classified as a separate genotype in the MLST method. Four of the strains classified to the largest genotype of BOX-PCR (3, 49, 53, and 57) also exhibited a high level of similarity in the *fliC* RFLP-PCR—above 98%; however, they were classified to three different genotypes (two isolates with numbers 53 and 57 recovered from obstetric ward) formed one cluster. The other three strains from the largest cluster of BOX-PCR method (23, 47 and 64) were classified by RFLP-PCR into one cluster counting four strains (with isolate 46); isolate 65 formed a separate genotype. In the MLST method, all these eight strains (namely 3, 23, 47, 49, 53, 57, 64 and 65) were classified as unique clusters consisting of only one strain. Four strains (1, 2, 24 and 39) have been found as members of the genotypes consisted of five isolates in the *fliC* RFLP-PCR and MLST methods. Isolates assigned with numbers 54 and 29 complemented these genotypes, respectively. However, more detailed analysis of both dendrograms revealed that in fact strains 54 and 29 exhibited a high level of similarity to those genotypes consisting of 5 isolates. Three of these strains (24 and 39, isolated in the unit of pediatrics in 2008; and one isolated in the internal ward of the hospital in 2007) were classified into one genotype by the BOX-PCR method and formed one genotype in all three methods. According to the BOX-PCR method, strains 2 and 29 were identified as unique genotypes consisted of only one strain and strain 54 was found as a member of the cluster counting three strains with numbers 68 and 44 (no relationship between these strains was found in other methods). A high level of similarity (close to 99%) between the seven following strains (-7, -11, -14, -17, -26, -31 and -62) was observed by the *fliC* RFLP-PCR and MLST methods. However, according to analysis performed strictly with borderline 99% similarity, these strains were divided into three different genotypes in *fliC* RFLP-PCR fingerprinting (one counting five strains (-7, -11, -14, -17, -26) and two clades consisted of separate strains, namely, 31 and 62) and three genotypes in the MLST method (two groups counting three strains (-7, -11, -14 and -17, -26, -31, respectively) and one separate strain, -62)). Interestingly, according to BOX-PCR, importantly different strains -11, -31 and -62 have been classified as separate genotypes, and isolates 17 and 26 formed a cluster consisting of two strains. Isolates 7 and 14 were recognized as members of the cluster consisting of four strains (together with 28 and 35); no important similarity between these strains have been confirmed in other methods. Strain 28 exhibited similarity with isolates 5 and 36 (in MLST, all of them belonged to one genotype, and in *fliC* RFLP-PCR similarity was about 98%). Strains 5 and 36 were classified into one genotype in all three methods, and both of them were isolated in 2008 in pediatrics unit. Three strains, 23, 47 and 64, were classified into one genotype in the *fliC* RFLP-PCR method, with one additional strain assigned with number 46. In MLST, all these strains were found as separate genotypes (with borderline of similarity 99%) but more detailed analysis of dendrogram revealed the high similarity between these strains (about 95%). The same method of differentiation suggested a high level of similarity between strains 55, 67, and 68; however, the results of RFLP-PCR and BOX typing revealed that these strains should be classified as separate genotypes. No other important relationships between the classifications of strains tested with the developed methods of genotyping have been found. 

Finally we compared the discriminatory power of the newly developed MLST method with the currently recommended approaches for genotyping of *E. coli* isolates: the Achman scheme [26], the Pasteur scheme [27] and the method developed by Clermond [28]. The results of in silico differentiation of the 15 whole genome sequences* (that were used for development of the new MLST method) with all four methods are presented in Table 1. The outcomes of this analysis are very optimistic for our MLST method. With this method the strains tested (n = 15) were classified to 13 different genotypes. Comparable results were obtained for Pasteur and Achtman schemes, in these methods 13 and 12 different genotypes were identified, respectively. The last of recommend methods, namely, the Clermond protocols, exhibited lower discriminatory power with seven genotypes. Only detailed analysis of the sequences of whole genomes (performed according to Kotlowski (2017) [29]) exhibited a higher potential of differentiation as it was expected the strains were classified to 15 different genotypes (Table 1). The dendrograms for all these methods are presented in the Appendix A (Appendix A).

From the epidemiology point of view, the outcomes of our investigation revealed the high genetic diversity of the strains tested. It seems that in most cases, the patients were infected with unique strains, probably from environmental sources—taking into account the results of all three methods, 46 genotypes are represented by only one strain. The remaining 25 strains can be divided into eight genotypes counting from two to seven isolates (Table 2). On the other hand, it is highly possible that some strains were transferred between patients. Classification to these eight genotypes is based on three methods of differentiation (Appendix A). The results presented in Table 2 indicate particularly high possibility of strains transfer between patients hospitalized in the units of pediatrics in 2008 (genotypes I, II, IV, V, and VIII), neurology in 2008 (genotype 2) or internal in 2007 (genotype IV), but also between the wards of the hospital—genotypes I, IV, V, VI, and VII. 

## 4. Materials and Methods 

### 4.1. Bacterial Strains

The *E. coli* strains (Table 3) were obtained from District Hospital in Wołomin, Poland. *E. coli* strains were isolated from urine samples of patients with cystitis. These strains were derived from the urine samples of symptomatic patients, in which ≥10 ^5^ colony-forming units of *E. coli* per ml were found. The strains were collected from January 2007 through December 2008 from five groups of patients hospitalized in the internal, pediatric, neurological, obstetric (pregnant women), orthopedics and dialysis hospital units. *E. coli* identification was performed according to standard procedures by Koneman et al. (1997) [30]. The identification of the strains was confirmed by using the ID 32 E system (bioMérieux).

### 4.2. Isolation of Genomic DNA

The strains were cultured in LB liquid medium overnight at 37 °C with shaking at 180 rpm. Genomic bacterial DNA was extracted using the Genomic Mini AX Bacteria Spin kit (A&A Biotechnology, Gdynia, Poland), according to the manufacturer’s instruction. The samples were stored at 20 °C.

### 4.3. BOX-PCR

The isolates were fingerprinted by BOX-PCR using BOX primer (5′-CTA-CGG-CAA-GGC-GAC-GCT-GAC-G-3′). Amplification was performed with 2× PCR Mix Plus HGC (A&A Biotechnology, Gdynia, Poland) in a total reaction volume of 25 μL containing 40 pmol of primer and 1 μL of genomic template DNA. BOX-PCR typing was carried out according to Dombek et al. [15] with a slight modification using the MasterCycler Gradient (Eppendorf) according to the following conditions: initial denaturation at 95 °C for 3 min followed by 35 cycles of PCR consisting of denaturation at 94 °C for 1 min, annealing at 48 °C for 2 min, and extension at 72 °C for 1.5 min; in the last cycle, the extension time was 5 min. 

### 4.4. Designing of Primers for Flic RFLP-PCR

The primers were designed based on the results of multiple alignments of a set of reference sequences of the flagellin-coding genes by using the CLUSTAL W program [31]. The accession numbers of all these sequences (n = 53) are presented in Appendix A (Appendix A; in each case, the accession number from GenBank was preceded by the number of flagellar—H antigen, e.g., H40_AJ884568.1). The simulation of differentiation of restriction fragments length polymorphism, including a selection of enzymes, was performed with SQRestriction—bioinformatics software developed by Perez-Marquez (2014) [32]. The reference sequences were previously selected and deposited by Joensen et al. (2015) [17].

### 4.5. Flic PCR-RFLP Analysis

The PCR-RFLP technique was employed for the genotyping of *E. coli* strains. For *fliC* gene amplification, the following PCR primer sequences were used: FLIC1: 5′-GGT-CAG-GCG-ATT-GCT-AAC-CG-3′ and FLIC2: 5′-TTG-GAC-ACT-TCG-GTC-GCA-TAG-TC-3′. The PCR was performed in the MasterCycler Gradient (Eppendorf) with 2× PCR Mix Plus HGC (A&A Biotechnology, Gdynia, Poland) in a reaction volume of 25 μL containing 40 pmol of primers and 1 μL of genomic template DNA. The PCR conditions included initial denaturation of the DNA at 94 °C for 1 min, followed by 30 repetitive cycles consisting of the following steps: denaturation of DNA at 94 °C for 1 min, annealing at 69 °C for 1 min and DNA elongation at 72 °C for 1 min. Final elongation was performed at 72 °C for 5 min. PCR products were digested with *Rsa*I restriction endonuclease (Thermo Fischer Scientific, Waltham, MA, USA) according to the manufacturer’s instruction.

### 4.6. Multilocus Sequence Typing (MLST)

Seven PCR reactions were performed using the following pairs of primers: Ec1, Ec2; Ec2, Ec4; Ec5, Ec6; Ec7, Ec7; Ec9, Ec10; Ec11, Ec12; Ec13, Ec14. PCR primer sequences are listed in Table 4 [11]. The sequences were selected on the basis of bioinformatics analysis of sequences of 15 *E. coli* whole genomes: AE014075.1; CP000243.1; CP000468.1; NC_011742.1; NC_011601.1; CP000948.1; U00096.3; CP002729.1; CP001509.3; NC_012967.1; NC_017656.1; NC_002695.1; NC_011748.1; CP007442.1; CU928163.2 [11]. PCR reactions were performed in the MasterCycler Gradient (Eppendorf) with 2× PCR Mix Plus HGC (A&A Biotechnology, Gdansk, Poland) in a reaction volume of 25 μL containing 40 pmol of primers and 1 μL of genomic template DNA. The PCR conditions included an initial denaturation of the DNA at 94 °C for 1 min, followed by 30 repetitive cycles consisting of the following steps: denaturation of DNA at 94 °C for 1 min, annealing for 1 min and DNA elongation at 72 °C for 1 min. Final elongation was performed at 72 °C for 10 min. The optimal annealing temperature for each pair of primers was calculated using a tm calculator (Thermo Scientific Web Tools).

The characteristics of all seven regions of the genomes that were considered as molecular targets for genotyping of *E. coli* strains is presented in Table 5 (this analysis was performed on the basis of the sequence of CP041955.1, an EC2 chromosome).

### 4.7. Analysis of PCR Products

The PCR products were separated on 2% agarose gel for around 3 h at 5 V/cm and stained with Midori Green. GeneRuler™ 100 bp Plus DNA Ladder (Thermo Fischer Scientific) was used as the DNA standard. The bands were visualized using a Molecular Imager^®^ Gel Doc™ XR+ System (Bio-Rad Laboratories Inc., Irvine, CA, USA). The images were processed and the size of the base pairs of the PCR products was estimated with Image Lab Software version 5.2.1 (Bio-Rad Laboratories Inc.). The resulting fingerprints were analyzed by MVSP software version 3.21. Dendrograms were obtained using the nearest neighbor cluster analysis.

## 5. Conclusions

In recent years, significant progress has been observed in the development of new techniques for the genotyping of pathogenic microorganisms [22,25,33]. Pulsed-field gel electrophoresis (PFGE) is still considered as the gold standard. However, development and access to the next-generation sequencing techniques have caused the analysis of differences in DNA, including whole-genome sequencing (WGS), to gain popularity [34,35,36] and it can be assumed that in the near future, these techniques will replace PFGE. Both these approaches (PFGE and sequencing), although guaranteed to obtain accurate and unequivocal results (especially in the case of sequencing), are expensive and time-consuming and their performance requires access to the specific equipment and the analysis of large sequences of DNA (whole genomes) remains an important bioinformatics challenge. Summarizing these techniques are available only for well-equipped laboratories and highly qualified personnel. For analysis of local outbreaks, where a limited number of strains are investigated, less advanced techniques are still used and the development of inexpensive, easy and fast-performing methods characterized with high discriminatory power of microorganisms genotyping is required. In our opinion, the herein presented methods, particularly MLST, meet all these expectations and can be recommended for the investigation of genetically diverse *E. coli* isolates in all laboratories.

## Figures and Tables

**Figure 1 pathogens-09-00073-f001:**
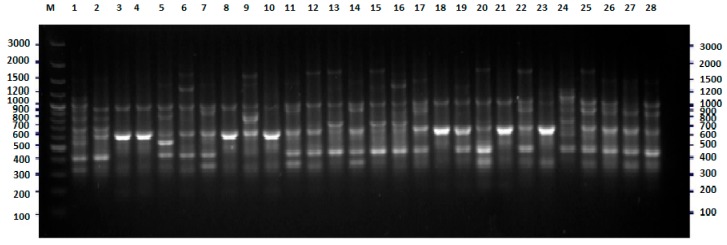
BOX-PCR fingerprinting of *E. coli* strains tested, the example of results for 28 isolates. The numbers indicate the numbers of particular isolates. M: 100 bp DNA ladder.

**Figure 2 pathogens-09-00073-f002:**
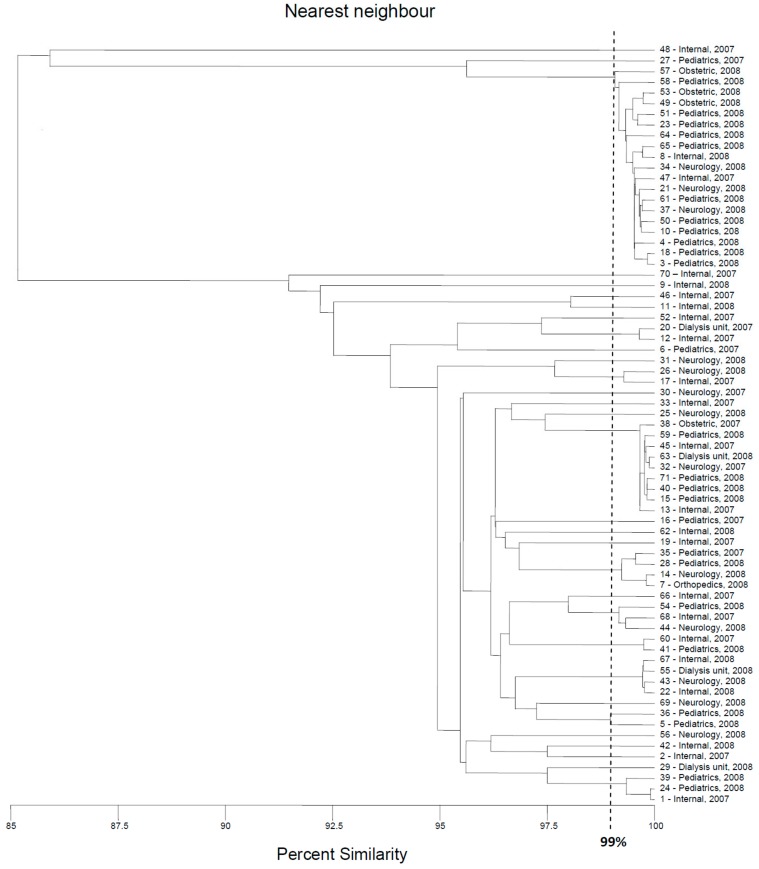
Dendrogram based on the nearest neighbor cluster analysis (MVSP software) of BOX-PCR fingerprints.

**Figure 3 pathogens-09-00073-f003:**
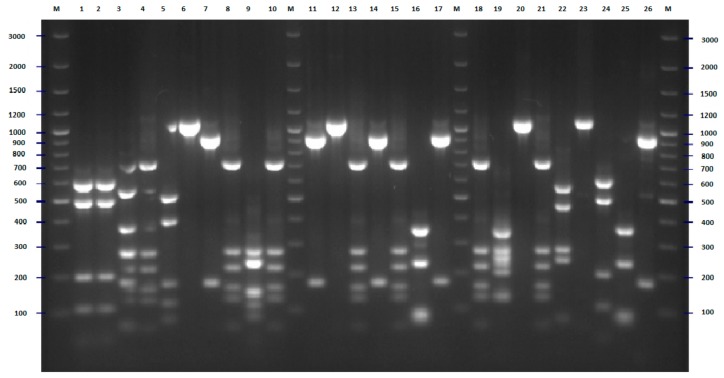
PCR-RFLP (polymerase chain reaction-restriction fragment length polymorphism) *Rsa*I restriction patterns of amplified *fliC* genes of *E. coli* strains tested—example of results for 26 isolates. The numbers indicate the numbers of particular isolates. M: 100 bp DNA ladder.

**Figure 4 pathogens-09-00073-f004:**
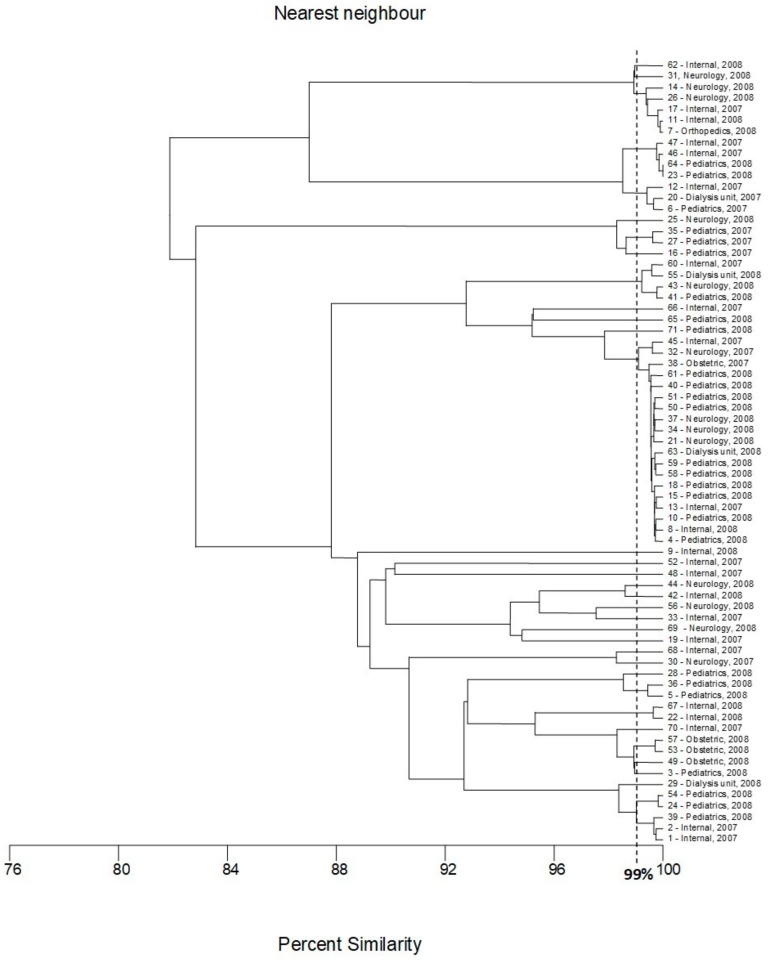
Dendrogram based on the nearest neighbor cluster analysis (MVSP software) of *fliC* RFLP-PCR fingerprints.

**Figure 5 pathogens-09-00073-f005:**
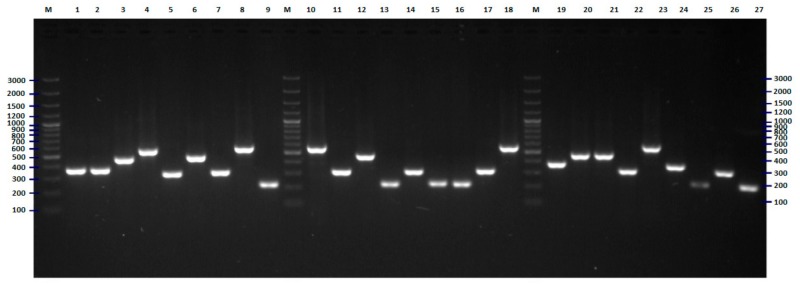
Multilocus sequence typing-polymerase chain reaction (MLST-PCR) fingerprinting of *E. coli* strains with primers Ec1, Ec2 (Table 4)—example of results for 27 isolates. The numbers indicate the numbers of particular isolates. M: 100 bp DNA ladder.

**Figure 6 pathogens-09-00073-f006:**
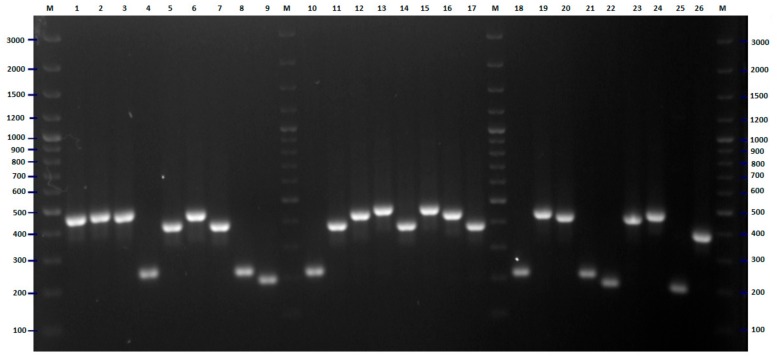
MLST-PCR fingerprinting of *E. coli* strains with primers Ec5, Ec6 (Table 4)—example of results for 26 isolates. The numbers indicate the numbers of particular isolates. M: 100 bp DNA ladder.

**Figure 7 pathogens-09-00073-f007:**
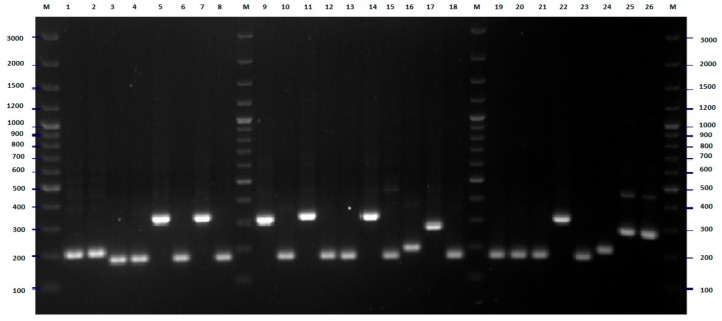
MLST-PCR fingerprinting of *E. coli* strains with primers Ec9, Ec10 (Table 4)—example of results for 26 isolates. The numbers indicate the numbers of particular isolates. M: 100 bp DNA ladder.

**Figure 8 pathogens-09-00073-f008:**
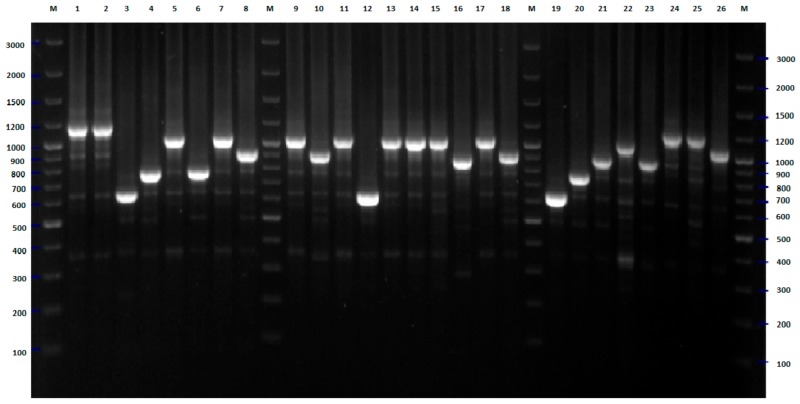
MLST-PCR fingerprinting of *E. coli* strains with primers Ec11, Ec12 (Table 4)—example of results for 26 isolates. The numbers indicate the numbers of particular isolates. M: 100 bp DNA ladder.

**Figure 9 pathogens-09-00073-f009:**
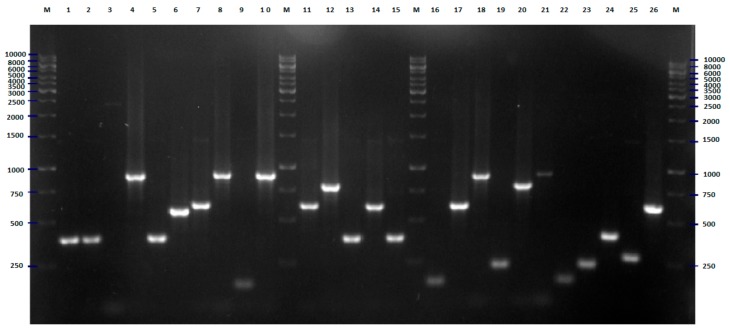
MLST-PCR fingerprinting of *E. coli* strains with primers Ec13, Ec14 (Table 4)—example of results for 26 isolates. The numbers indicate the numbers of particular isolates. M: 250 bp DNA ladder.

**Figure 10 pathogens-09-00073-f010:**
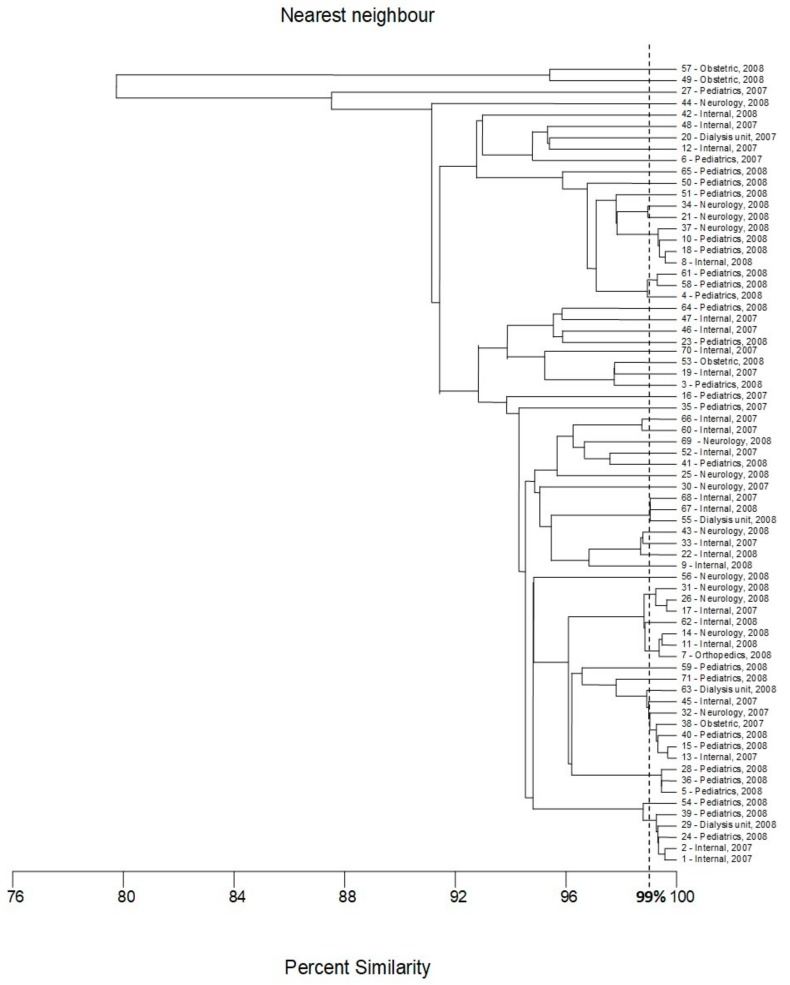
Dendrogram based on the nearest neighbor cluster analysis (MVSP software) of MLST fingerprints.

**Table 1 pathogens-09-00073-t001:** Comparison of four reference methods with the new PCR-MLST method.

	Genotyping Method	The Lowest Similarity between Strains Tested	Genotypes Number n = 15 *
1	Genomes similarity—according Kotlowski (2017) [29]	91.15	15
*2*	*E. coli* #1; MLST (*adk; fumC; gyrB; icd; mdh; purA; recA*)— Achtman scheme [26]	99.63	12
*3*	*E. coli* #2; MLST (*dinB; icdA; pabB; polB; putP; trpA; trpB; uidA*)— Pasteur scheme [27]	94.50	13
4	Genotyping by Clermont O. for 7 genomic regions [28]	63.78	7
5	New PCR-MLST method—this study	80.86	13

* *E. coli* nissle probiotic strain was excluded from the analysis.

**Table 2 pathogens-09-00073-t002:** Classification of the strains tested to genotypes, taking into account results of all three methods (BOX-PCR, RFLP-PCR, and MLST) *.

Number of Genotype	Number of Strains Classified to the Genotype	Numbers of Strains Classified to the Genotype	Ward of Hospital	Year of Isolation	Gender of Patient
I	4	8	Internal	2008	Female
		10	Pediatrics	2008	Female
		18	Pediatrics	2008	Female
		37	Neurology	2008	Female
II	3	4	Pediatrics	2008	Male
		58	Pediatrics	2008	Female
		61	Pediatrics	2008	Female
III	2	21	Neurology	2008	Female
		34	Neurology	2008	Male
IV	7	13	Internal	2007	Male
		15	Pediatrics	2008	Male
		32	Neurology	2007	Female
		38	Obstetric	2007	Female
		40	Pediatrics	2008	Female
		45	Internal	2007	Female
		63	Dialysis unit	2008	Female
V	3	1	Internal	2007	Female
		24	Pediatrics	2008	Male
		39	Pediatrics	2008	Male
VI	2	7	Orthopedics	2008	Female
		14	Neurology	2008	Unknown
VII	2	17	Internal	2008	Female
		26	Neurology	2008	Female
VIII	2	5	Pediatrics	2008	Female
		36	Pediatrics	2008	Female

* Other isolates (n = 46) were classified to genotypes consisted of only one strain.

**Table 3 pathogens-09-00073-t003:** *E. coli* strains used for the study.

No.	*E. coli* Strains	Ward of Hospital	Year of Isolation	Gender of Patient
1	34	Internal	2007	Female
2	14653	Internal	2007	Female
3	2878	Pediatrics	2008	Female
4	6471	Pediatrics	2008	Male
5	13202	Pediatrics	2008	Female
6	13917	Pediatrics	2007	Male
7	66	Orthopedics	2008	Female
8	2937	Internal	2008	Female
9	7271	Internal	2008	Male
10	7308	Pediatrics	2008	Female
11	10492	Internal	2008	Female
12	15193	Internal	2007	Female
13	14294	Internal	2007	Male
14	863	Neurology	2008	Unknown
15	13356	Pediatrics	2008	Male
16	11199	Pediatrics	2007	Male
17	7242	Internal	2008	Female
18	6897	Pediatrics	2008	Female
19	2267	Internal	2007	Female
20	1757	Dialysis unit	2007	Female
21	14543	Neurology	2008	Female
22	9952	Internal	2008	Female
23	456	Pediatrics	2008	Male
24	4838	Pediatrics	2008	Male
25	9478	Neurology	2008	Female
26	1694	Neurology	2008	Female
27	2648	Pediatrics	2007	Female
28	7663	Pediatrics	2008	Female
29	6780	Dialysis unit	2008	Female
30	13583	Neurology	2007	Female
31	124	Neurology	2008	Female
32	13192	Neurology	2007	Female
33	8784	Internal	2007	Male
34	4187	Neurology	2008	Male
35	8956	Pediatrics	2007	Male
36	10316	Pediatrics	2008	Female
37	14158	Neurology	2008	Female
38	12921	Obstetric	2007	Female
39	8851	Pediatrics	2008	Male
40	9683	Pediatrics	2008	Female
41	10238	Pediatrics	2008	Female
42	9010	Internal	2008	Female
43	6429	Neurology	2008	Female
44	13567	Neurology	2007	Female
45	2421	Internal	2007	Female
46	1368	Internal	2007	Male
47	12021	Internal	2007	Female
48	9898	Internal	2007	Male
49	5326	Obstetric	2008	Female
50	5660	Pediatrics	2008	Male
51	10444	Pediatrics	2008	Female
52	4666	Internal	2007	Female
53	3850	Obstetric	2008	Female
54	10136	Pediatrics	2008	Female
55	947	Dialysis unit	2008	Female
56	7522	Neurology	2008	Male
57	5326	Obstetric	2008	Female
58	2310	Pediatrics	2008	Female
59	7133	Pediatrics	2008	Male
60	1908	Internal	2007	Female
61	4410	Pediatrics	2008	Female
62	1036	Internal	2008	Female
63	8677	Dialysis unit	2008	Female
64	12497	Pediatrics	2008	Male
65	3909	Pediatrics	2008	Female
66	11764	Internal	2007	Female
67	4500	Internal	2008	Female
68	3039	Internal	2007	Female
69	6114	Neurology	2008	Male
70	5165	Internal	2007	Female
71	9046	Pediatrics	2008	Male

**Table 4 pathogens-09-00073-t004:** Multilocus sequence typing (MLST) PCR primers sequences [11] and their location within the chromosome of one of the reference strains of *E. coli* (CP041955.1, EC2 chromosome).

No.	Name	PCR Primer Sequences (5′–3′)	Location in CP041955.1, EC2 Chromosome
1	Ec1	5′-ATC-GGC-CAT-ATC-AAG-TCG-ATG-TTG-TTG-CA-3′	3265597-3265625
	Ec2	5′-TGC-TTA-CTT-CGC-CGT-GGA-TAC-TAC-3′	3265309-3265332
2	Ec3	5′-AGA-AGC-AGC-TGC-AGC-TGA-AGC-AAG-A-3′	3246933-3246957
	Ec4	5′-GCT-GCT-TTC-TTA-TCA-GCT-GCT-GCC-T-3′	3246763-3246787
3	Ec5	5′-GCC-GAA-CGT-TCA-CCA-CTA-CAA-AAG-3′	1578518-1578541
	Ec6	5′-CAG-ATC-GGC-TTC-GCT-TAG-CTG-3′	1578158-1578178
4	Ec7	5′-GTT-CAG-TGT-TTC-AAT-TTT-CAG-CTT-GAT-CCA-G-3′	3835984-3836014
	Ec8	5′-TGC-GGT-TGG-ATC-ACC-TCC-TTA-CCT-3′	3836352-3836375
5	Ec9	5′-GAA-ATT-ACG-CAA-GAT-TCG-CTG-GTG-CA-3′	154450-154475
	Ec10	5′-GGA-ATT-TCT-GGT-TCG-TAA-AGT-ACG-ATG-3′	154157-154183
6	Ec11	5′-GAA-GGA-GAA-GAA-AAT-TCA-GGA-AAT-GGA-TAA-AG-3′	1153234-1153265
	Ec12	5′-GCT-AAG-GGA-GTA-TGC-GGT-CAA-AAG-3′	1152225-1152248
7	Ec13	5′-GCG-GAT-CAA-TAA-TTG-AAG-ATT-GCCG-GGG-3′	4412457-4412484
	Ec14	5′-CTG-AGC-TGT-ATG-GAA-GGG-TTT-GAT-ACC-G-3′	4412332-4412359

**Table 5 pathogens-09-00073-t005:** Location of target sequences within CP041955.1, an EC2 chromosome.

Target Number	Amplified Region of the Genome	
1.	Succinate dehydrogenase iron-sulfur subunit, QGQ16019.1	
2.	Cell envelope integrity protein TolA, QGQ16014.1	
3.	Hypothetical protein QGQ14586.1, ISNCY family transposase QGQ14587.1	
4. *	Locus_tag = “FOZ67_18655”/product = “23S ribosomal RNA”	Locus_tag = “FOZ67_18670”/product = “16S ribosomal RNA”.
5. *	Product = “tRNA uridine(34)/cytosine(34)/5-carboxymethylaminomethyluridine(34)-2′-O)-methyltransferase TrmL”/protein_id = “QGQ13328.1”	Gene “lldD”/product= “quinone-dependent L-lactate dehydrogenase”/protein_id = “QGQ13329.1”
6. *	Locus_tag = “FOZ67_05640” tRNA/locus_tag = “FOZ67_05640”/product = “tRNA-Ser”	Gene = “yqaB”/locus_tag = “FOZ67_05665”
7. *	Locus_tag = "FOZ67_21460"	Sel1 repeat family protein"/protein_id = "QGQ17016.1".

* Two columns indicate the location between target genes.

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
