# Peer review of "New Approaches for Escherichia coli Genotyping"

_pathogens, 2020, doi:10.3390/pathogens9020073_

Round 1

Reviewer 1 Report

The manuscript entitled “New approaches for Escherichia coli genotyping” aimed at proposing two new, original tools, fliC RFLP-PCR and MLST, that can be useful for genotyping of E. coli isolates. The authors demonstrated that both methods exhibited comparable (RFLP- PCR) or higher (MLST) potential for differentiation of the strains tested in comparison to BOX-PCR method. Overall, in this work, the presenting data can suitably support their conclusions. I have some points associated with this manuscript.

Line 67: “chronic diarrh0ea” was incorrectly written as “chronic diarrhea". The number of selected genes for MLST is not consistent in Line 77 and Line 149. Please spell small numbers out. The small numbers, such as whole numbers smaller than ten, should be spelled out (Line 96, Line 104, Line 149, Line 220 …). The author should quantify cladogram topology using well-established topology metrics (e.g., Robinson-Foulds).

Author Response

Reviewer 1

We are very grateful for all comments and remarks. Below we have presented detailed explanations to all questions and remarks. To the manuscript we have also introduced some other changes – according to suggestions of other reviewers. Moreover we have changed the order of the authors.  

Line 67: “chronic diarrh0ea” was incorrectly written as “chronic diarrhea". – we are very sorry for this error, it has been changed.

The number of selected genes for MLST is not consistent in Line 77 and Line 149 – of course, the reviewer is right, the information in line 77 has been changed, we used 7 genes for analysis.

Please spell small numbers out. The small numbers, such as whole numbers smaller than ten, should be spelled out (Line 96, Line 104, Line 149, Line 220 …) – has been changed

The author should quantify cladogram topology using well-established topology metrics (e.g., Robinson-Foulds). 

For our analysis we used nearest neighbour cluster analysis (MVSP software) – all dendrograms presented in the manuscript are based on this method, which is also commonly used by other authors.

In this method, a similarity matrix (in percentage) of all strains tested (each with each) is prepared. The dendrogam is created on the basis of the results presented in this matrix.  We set a 99% similarity as the borderline to classification of two strains to the same genotype. This is a very high value, but it guarantees that strains classified to the same cluster are identical - they come from the same source. We would be grateful if you could accept this explanation and the form of presentation of the results of our studies. In our opinion it should be clear for potential readers.

Reviewer 2 Report

In the manuscript New approaches for Escherichia coli genotyping authors developed two methods for typing of Escherichia coli isolates. First method, a PCR-RFLP, is based on PCR amplification of variable fliC gene and second method is multi-locus sequence typing. The developed methods and their discriminatory power were tested on the set of 71 uropathogenic E. coli isolates.

 The manuscript is well-written, but I have fundamental concerns with respect to reasons and needs for this study, experimental design, and obtained outputs and conclusions.

Major comments:

Authors developed two new methods for genotyping of coli, however several different genotyping methods exist at this moment and they are broadly and successfully used in basic research and epidemiology studies around the world. There are genotyping methods based on serology (serotyping of O:K:H antigens), restriction profiling (PFGE, RFLP, Rep_PCR, etc.), and sequencing (MLST). While some of them are listed in the manuscript (BOX-PCR), crucial information about well-known PCR-phylotyping (Clermont et al 2000 and 20013; analysis of 3 and 7 loci) and “Achtman scheme” and “Pasteur scheme” for MLST (analysis of 6 and 8 loci respectively) are omitted. Besides low needs for developing of new genotyping methods, insufficient information about developed methods is second important issue of this study. According to methodology, there are no information about loci/regions/genes, which were used for MLST. The authors referenced to their previous publication, but it is only a short comment instead of full research article. In addition, there are not shown sequence numbers for complete coli genomes and also for fliC gene, which were used for developing of new methods, MLST and RFLP, respectively. As a “reference method”, authors performed a BOX-PCR (Dombek et al. 2000); however, they performed important modifications in protocol (i.e., synthesis 1 min instead of 8 min), which has significant effect on obtained fingerprints. Moreover, presentation of obtained results (gel electrophoresis) is not optimal (i.e. smears instead of discrete bands in Fig. 1). The comparison of developed MLST with other already existing MLST systems (i.e., Achtman/Pasteur) will be more suitable than with BOX-PCR analysis. Authors presented that the developed techniques have same or even better discriminatory power than BOX-PCR performed with their modification. Out of 71 UPEC strains/isolates, 25 coli were classified to one of the 8 genotypes (based on BOX-PCR, fliC and MLST) and 46 isolates were unique. However, the manuscript completely omitted the information/analysis how newly developed genotyping reflect real phylogenesis of E. coli strains. Moreover, how variability in analyzed loci (i.e., fliC and seven uncharacterized loci of MLST) reflects the variability on a genomic level? Without these analyses and without direct comparisons of new methods with well-known typing techniques, the new genotyping methods are of limited value. The manuscript should be substantially shortened, especially discussion repeating results. Figures with gels should be completed by schemes of principles of methods. Table 2 should be presented in the supplementary material.

Author Response

Reviewer 2

We are very grateful for all comments and remarks. Below we have presented detailed explanations to all questions and remarks. To the manuscript we have also introduced some other changes – according to suggestions of other reviewers. Moreover we have changed the order of the authors.  

In the manuscript “New approaches for Escherichia coli genotyping” authors developed two methods for typing of Escherichia coli isolates. First method, a PCR-RFLP, is based on PCR amplification of variable fliC gene and second method is multi-locus sequence typing. The developed methods and their discriminatory power were tested on the set of 71 uropathogenic E. coli isolates.

 The manuscript is well-written, but I have fundamental concerns with respect to reasons and needs for this study, experimental design, and obtained outputs and conclusions.

Major comments:

Authors developed two new methods for genotyping of coli, however several different genotyping methods exist at this moment and they are broadly and successfully used in basic research and epidemiology studies around the world. There are genotyping methods based on serology (serotyping of O:K:H antigens), restriction profiling (PFGE, RFLP, Rep_PCR, etc.), and sequencing (MLST). While some of them are listed in the manuscript (BOX-PCR), crucial information about well-known PCR-phylotyping (Clermont et al 2000 and 20013; analysis of 3 and 7 loci) and “Achtman scheme” and “Pasteur scheme” for MLST (analysis of 6 and 8 loci respectively) are omitted.

We are very grateful to the reviewer for this comment. The excellent methods – Achtman scheme and Pasteur scheme are taken into account in the new version of the manuscript. However, we would not fully agree with the opinion that there is a low need for developing of new genotyping methods. The mentioned above methods are based on sequencing of selected fragments of the genomes – thus these methods are expensive. Both our method are based on analysis of sizes of amplified fragments of DNA, as a consequence the cost of analysis is importantly lower. It was also the reason why BOX-PCR method was selected as a reference.

Besides low needs for developing of new genotyping methods, insufficient information about developed methods is second important issue of this study. According to methodology, there are no information about loci/regions/genes, which were used for MLST. The authors referenced to their previous publication, but it is only a short comment instead of full research article. In addition, there are not shown sequence numbers for completecoli genomes and also for fliC gene, which were used for developing of new methods, MLST and RFLP, respectively.

We are grateful for this comment. We have added the location of the primers’ sequences within genome of one of reference strains of E. coli that was used for bioinformatics analysis – it is presented in the Table 4. The numbers of sequences of whole genomes are added to the section – materials and methods. The numbers of sequences of all fliC genes (these are 53 sequences) are presented in supplementary materials (Figures S1 and S2) – a short comment is added to the section Materials and methods.       

As a “reference method”, authors performed a BOX-PCR (Dombek et al. 2000); however, they performed important modifications in protocol (i.e., synthesis 1 min instead of 8 min), which has significant effect on obtained fingerprints. Moreover, presentation of obtained results (gel electrophoresis) is not optimal (i.e. smears instead of discrete bands in Fig. 1).

I have to admit that we optimized BOX-PCR in our laboratory, and we did not obtain better results (in term of both – quality of results e.g. presence of smears as well as discriminatory power) using the method strictly described by Dombek et al. (with 8 min of synthesis). Thus we decided to use the modified protocol with 1 min of synthesis. In my opinion other factors, e.g. the quality of PCR polymerase or buffer components used for PCR amplification are crucial for the quality of obtained results (is more important than the time of synthesis).

The comparison of developed MLST with other already existing MLST systems (i.e., Achtman/Pasteur) will be more suitable than with BOX-PCR analysis. Authors presented that the developed techniques have same or even better discriminatory power than BOX-PCR performed with their modification. Out of 71 UPEC strains/isolates, 25 coli were classified to one of the 8 genotypes (based on BOX-PCR, fliC and MLST) and 46 isolates were unique. However, the manuscript completely omitted the information/analysis how newly developed genotyping reflect real phylogenesis of E. coli strains. Moreover, how variability in analyzed loci (i.e., fliC and seven uncharacterized loci of MLST) reflects the variability on a genomic level? Without these analyses and without direct comparisons of new methods with well-known typing techniques, the new genotyping methods are of limited value.

In accordance with the reviewer's recommendations, we conducted a bioinformatics analysis of differentiation of 15 reference strains of E. coli with known sequences of whole genomes (AE014075.1; CP000243.1; CP000468.1; NC_011742.1; NC_011601.1; CP000948.1; U00096.3; CP002729.1; CP001509.3; NC_012967.1; NC_017656.1; NC_002695.1; NC_011748.1; CP007442.1; CU928163.2;) with Achtman, Pasteur, Clermond and proposed in our manuscript new MLST method. Additionally analysis of whole genomes (according to method described by Kotlowski 2017) was performed. The aim of this analysis was comparison of differentiation power of these genotyping methods. The results of are presented in Table 1 (new element of the manuscript). The outcomes of this analysis are very optimistic for our MLST method. With this method the strains tested (n=15) were classified to 13 different genotypes. The same result was obtained Pasteur protocol, other recommend methods, namely Achtman and Clermond protocols, exhibited lower discriminatory power with 8 and 7 genotypes respectively. Except of the Table we also added a short description of these results. The most important advantage of fliC RFLP-PCR method is the fact that it can be used insted of H serotyping of E. coli isolates. However this method also exhibits high discriminatory power and can be applied for genotyping of E. coli strains.

One general comment – the E. coli nissle probiotic strain is excluded from this analysis (thus 15 not 16 whole genome sequences of E. coli were used for bioinformatics analysis – in this part of investigation)

The manuscript should be substantially shortened, especially discussion repeating results. Figures with gels should be completed by schemes of principles of methods. Table 2 should be presented in the supplementary material. 

We also considered moving Table 2. However, many other information is presented in the supplementary materials, and it can be not convenient for potential readers. If only it is possible we would like to leave this Table in the section Materials and methods. We also would not like to remove any information from discussion. In this part of the manuscript we wanted to highlight advantages of both new methods (other reviewers did not suggest that it is necessary to shorten the manuscript).  

Reviewer 3 Report

Here the authors present two novel genotyping methods for clinical strains of E. coli. They developed these methods to meet the need of cost effective, reproducible and simple typing method. The first method was based on typing the fliC gene which encodes the H antigen and the second method was an amended protocol based on MLST typing. These methods were compared to the reference BOX-PCR method. Each of the three methods differed in their level of discrimination, regardless the novel methods were able to identify strains with a likely epidemiological significance.

General Comments

Spelling Errors:

Line 67: Please amend diarrh0ea to diarrhea (USA) or diarrhoea (English)

Line85: Please amend et to at

Line 111: There needs to be a close bracket ) after S1. 

Line 224: Please amend tents to tens

Line 295: Should ‘no’ be ‘number’?

Line 343: Please amend MLTS to MLST

The following need to be in italics:

Line 82: E. coli

Line 84: E. coli

Line 107: filC

Line 112: RsaI

Line 133: fliC

Line 135: in silico

Line 139: filC

Line 142: RsaI, filC and E. coli

Line 146: filC

Line 181: E. coli

Line 184: E. coli

Line 187: E. coli

Line 190: E. coli

Line 193: E. coli

Line 353: et al

Line 365: et al

Line 375: et al

Line 88: Should this be two sentences? ‘The largest cluster contained 19 strains, almost all isolated in 2008. Eleven, three, three and two of them were recovered from patients of the pediatric, obstetric, neurology and internal units, respectively.’

Line 140: I could not find Figure S4 in the material supplied to me.

Specific Comments

Regarding the MLST typing methodology, the sizes of the base pairs were determined using imaging software. However, you state that these methods do not require specialised equipment. Can you comment on the specificity/accuracy of determining band size and how this might be effected if you do not have access to imaging software? If you size the bands incorrectly this will effect the clustering of the strains. I feel this is a limitation which has been overlooked. 

Can you mention how these typing methods identify novel strains which have not been previously typed? 

As both the filC and MLST methods do not use a curated database of E. coli strains how would these methods be useful in a larger outbreak, for example across multiple healthcare settings or longitudinally to track specific strains? 

The discrimination of each of the methods varies. Is than an advantage regarding pathogenicity to type using one method over another? 

Author Response

Reviewer 3

First of all we are very grateful for all comments and remarks. We are also very grateful for indication of some typing errors in the text of the manuscript – all of them have been taken into account and changed in the text (all changes are marked). Below we have also presented detailed explanations to all questions and remarks. To the manuscript we have also introduced some other changes – according to suggestions of other reviewers. Moreover we have changed the order of the authors.  

Comments and Suggestions for Authors

Here the authors present two novel genotyping methods for clinical strains of E. coli. They developed these methods to meet the need of cost effective, reproducible and simple typing method. The first method was based on typing the fliC gene which encodes the H antigen and the second method was an amended protocol based on MLST typing. These methods were compared to the reference BOX-PCR method. Each of the three methods differed in their level of discrimination, regardless the novel methods were able to identify strains with a likely epidemiological significance.

General Comments – all these changes are introduced to the manuscript

Spelling Errors:

Line 67: Please amend diarrh0ea to diarrhea (USA) or diarrhoea (English)

Line85: Please amend et to at

Line 111: There needs to be a close bracket ) after S1. 

Line 224: Please amend tents to tens

Line 295: Should ‘no’ be ‘number’?

Line 343: Please amend MLTS to MLST

The following need to be in italics:

Line 82: E. coli

Line 84: E. coli

Line 107: filC

Line 112: RsaI

Line 133: fliC

Line 135: in silico

Line 139: filC

Line 142: RsaI, filC and E. coli

Line 146: filC

Line 181: E. coli

Line 184: E. coli

Line 187: E. coli

Line 190: E. coli

Line 193: E. coli

Line 353: et al

Line 365: et al

Line 375: et al

Line 88: Should this be two sentences? ‘The largest cluster contained 19 strains, almost all isolated in 2008. Eleven, three, three and two of them were recovered from patients of the pediatric, obstetric, neurology and internal units, respectively.’

We agree with the reviewer’s opinion – this fragment has been changed; now there are two sentences.

Line 140: I could not find Figure S4 in the material supplied to me. – should be S3 it is changed

Specific Comments

Regarding the MLST typing methodology, the sizes of the base pairs were determined using imaging software. However, you state that these methods do not require specialised equipment. Can you comment on the specificity/accuracy of determining band size and how this might be effected if you do not have access to imaging software? If you size the bands incorrectly this will effect the clustering of the strains. I feel this is a limitation which has been overlooked. 

We have an access to the specialised equipment (and software) for gel analysis and we used them in our study. Of course, we agree that using this equipment guarantees obtaining more accurate and unambiguous results. It is also very convenient and analysis can be done in a shorter period of time.  However, in our opinion the same analysis (at the same, or at least comparable, level of accuracy) could be done without this software/equipment – using only visual assessment of DNA fragments’ size (both, PCR products or products of DNA digestion with restriction enzyme). In fact this equipment is new in our laboratory, and from our previous experience we can say that we can do assessment of size of DNA fragment with the accuracy of +/- 10 bps (at least for smaller DNA molecules below 600 - 500 bp). In the case of fliC RFLP-PCR method as a result different patterns of DNA fragments (usually below 500 bps) are produced, and in our opinion visual analysis is completely enough for classification of different genotypes. We have highlighted this fact – as an important advantage (lines 114-115) of this method.

In the case of MLST method five different loci are taken into account, and for three of them (amplified with pairs of primers: Ec1-Ec2, Ec5-Ec6 and Ec9-Ec10) the amplified fragments of DNA are at about 500 - 600 bps or less – thus, visual analysis should be sufficient. For two other targets (amplified with pairs of primers Ec11-Ec12 and Ec13-Ec14) larger fragments of DNA are amplified (for some strains). However, in the case of both these targets our analysis revealed important differences in the size of amplified fragments and differentiation power (particularly in the case of PCR carried out with Ec13-Ec14). In this method (MLST) all five targets are taken into account for characterization of each strain tested and we do believe that visual assessment is completely sufficient for differentiation of investigated, even closely related E. coli strains. 

Can you mention how these typing methods identify novel strains which have not been previously typed? 

Both, proposed in our manuscript, methods are species specific – all five targets of MLST method and fliC gene are characteristic for E.coli. Thus, in our opinion each strain that belongs to the species of E. coli can be typed with both these methods.  Moreover, the designed primers are highly specific and PCR amplification can be performed using high temperature of annealing of primers – 69 °C (amplification of the fragment of fliC gene) and 72 °C (all five targets of MLST) – the probability of amplification of nonspecific targets (e.g. other fragments of E. coli genome or fragments of genomes of other bacteria) in these conditions is nearly completely eliminated.

As both the filC and MLST methods do not use a curated database of E. coli strains how would these methods be useful in a larger outbreak, for example across multiple healthcare settings or longitudinally to track specific strains? 

In our manuscript we have proposed two new methods of genotyping of E. coli. In both these methods specific targets are amplified (as it is explained above probability amplification of artificial PCR products is eliminated), thus we can easily compare (on the basis of DNA bands patterns in the case of fliC RFLP-PCR or size of all five targets in the case of MLST method) strains from different places (e.g. in the case of multiple healthcare settings) or track for specific strains. We believe that in the future for our methods some databases of strains of E. coli that characterize with specific combinations of DNA fragments (of determined size) will be created. We are also going to make some effort in this area. 

The discrimination of each of the methods varies. Is than an advantage regarding pathogenicity to type using one method over another? 

In general, MLST method characterize with higher discriminatory power. However, we do not think that there exist any link between pathogenicity and classification of the strains to particular genotypes. However, we would like to notice that fliC RFLP-PCR can be used instead of serotyping of H flagellar antigen. There is a correlation between pathogenicity of E. coli and type of H antigen produced in the cells. Thus in the future the band patterns which are characteristic for strains of higher pathogenicity could be recognized and selected as a reference for the identification of E. coli pathogenic strains. 

Round 2

Reviewer 2 Report

The authors responded to my previous comments and suggestions, but not completely. Careful reading of the revised manuscript and authors answers did not convince me about obtained outputs of this manuscript and I have these comments.

Comments:

Add 1. Authors argue that their methods are cheaper than sequencing. Authors should support their statement by calculation. For sequencing, the samples are isolated by commercial kit and can be directly sequenced in commercial facilities, while isolation of DNA (same as for sequencing) and further PCR amplifications (1-5 reaction per sample) together with digestion by restriction enzymes are needed for their techniques. The purchase of PCR cycler, electrophoresis equipment and gel documentation software are quite expensive. Moreover, information obtained from sequencing is more comprehensive, than from PCR (only PCR product size).

Add 2. The authors added only genomic coordinates of primers used in MLST. But the information what is encoded in amplified regions is more important. Without the knowledge, what is used for genotyping, the novel methods are of the limited value. The authors should characterize these regions and also clarify why are these regions suitable for genotyping. For example, some regions should be represented by mobile genetic elements/prophages, which can significantly affect the genotyping outputs.

Add 3. Authors compared the discriminatory power of their novel methods with other known typing techniques. The novel techniques are able to distinguish various E. coli strains on the same or better level than generally used methods, but the authors did not show the cluster analysis for these various techniques. Are dendrograms similar or different for methods presented in Table 1? These results can confirm or not the relevance of genomic regions used for novel genotyping.

Add 4. Besides comparisons of dendrograms from various methods (see above), dendrograms presented in this manuscript should be completed with “fingerprints” (PCR profiles) used for constructions of these trees. For example, Fig. 1 and 2 should be combined and presented together. Moreover, the fingerprints of BOX-PCR have a poor quality and bands used in cluster analysis should be marked.

Add 4b. While a lot of results is presented in discussion (e.g. whole paragraph L335-389), there are not showed the fingerprints of genotypes identified in this study and presented in Table 2.  Without presenting of type-fingerprints for identified genotypes the obtained data will be not comparable with following studies. In this form, the results only showed that E. coli strains/isolates are different, but it is not possible to classify these E. coli isolates. Classification (based on similarity or phylogeny) is important part of genotyping.

Author Response

First of all we are very grateful for all comments and suggestions. The detailed answers are presented under the opinions of the reviewer.

Comments and Suggestions for Authors

The authors responded to my previous comments and suggestions, but not completely. Careful reading of the revised manuscript and authors answers did not convince me about obtained outputs of this manuscript and I have these comments.

Comments:

Add 1. Authors argue that their methods are cheaper than sequencing. Authors should support their statement by calculation. For sequencing, the samples are isolated by commercial kit and can be directly sequenced in commercial facilities, while isolation of DNA (same as for sequencing) and further PCR amplifications (1-5 reaction per sample) together with digestion by restriction enzymes are needed for their techniques. The purchase of PCR cycler, electrophoresis equipment and gel documentation software are quite expensive. Moreover, information obtained from sequencing is more comprehensive, than from PCR (only PCR product size).

First of all I agree with the reviewer that sequencing is more comprehensive and unequivocal than all genotyping methods that are based on PCR or even PFGE (Pulsed field gel electrophoresis). However, we still think that both methods proposed in our manuscript are importantly cheaper than sequencing.

We do not have equipment for sequencing and we do it as an external service. The cost for one sample (one read 800 bps.) is about 5 Euro (including our discount). For each gene two reads should be performed (with forward and reverse primers, respectively), which means that cost of sequencing of one target is about 10 Euros (these are the costs in our laboratory). Thus for Achtman scheme (six different targets) the cost of analysis of one strain would be 60 Euro – in our opinion it is quite high cost. Of course I realize that the costs can be lower for other institutes that have equipment for sequencing or have some special agreements with other institutes/laboratories who can perform sequencing for them as a service. The presented above calculation shows the situation in our laboratory (cost of analysis of one strain with Achtman scheme – about 60 Euro).

The costs of analysis with PCR include: 1) PCR amplification (the cost of MIX is only about 0.25 Euros per one reaction – in the volume of 25 µl. We buy the mix from A&A Biotechnology, Gdynia, Poland); 2) agarose electrophoresis (agarose, buffer for electrophoresis and DNA standards). For agarose electrophoresis we use a gel for 30 samples (including DNA ladder). For preparing the gel (300 ml 2% agarose) we use 6 grams of agarose and the cost of this material is about 0.5 Euro per gram. Thus, the cost of the whole gel is not higher than 3-5 Euro (including costs of buffer). At least 5 strains (all five targets proposed in our MLST scheme) can be investigated using one gel. This calculation clearly shows that cost of analysis with our MLST method (but also RFLP-PCR, only one additional step – digestion is necessary) is much lower in comparison to the methods that require sequencing (the cost is certainly much lower than 60 Euro per one strain). Of course cycler and agarose electrophoresis equipment are required, but in our opinion these devices are much more common than equipment for sequencing and are available even in not really well equipped laboratories. For both MLST and PCR-RFLP method presented in our manuscript any specific and expensive equipment or software for gel documentation or analysis is not strictly required – as we explained in the previous round of revision the analysis of sizes of PCR fragments can be performed visually (of course using software guarantee obtaining more unequivocal results).

Add 2. The authors added only genomic coordinates of primers used in MLST. But the information what is encoded in amplified regions is more important. Without the knowledge, what is used for genotyping, the novel methods are of the limited value. The authors should characterize these regions and also clarify why are these regions suitable for genotyping. For example, some regions should be represented by mobile genetic elements/prophages, which can significantly affect the genotyping outputs.

The selection of primers for MLST method is based on the in silico comparative analysis of sequences of 15 genomes of E. coli. This analysis revealed that sequences of these primers are present in all 15 genomes (our investigation additionally confirmed that they were present in all 81 strains tested). In our opinion important advantage of these primers is the fact that all of them are species specific. Moreover regions of genomes “located between” these primers importantly differ in term of sequence but also size – which can be easily determined by amplification of these fragments with PCR and agarose gel electrophoresis. The information about targets selected for differentiation of E. coli isolates are presented in Table 5. As shown in the Table in the case of four out of seven targets (targets 4-7) the amplified fragments are located between two coding genes, and probably these are noncoding intergenic spacer regions. In other mentioned MLTS methods dedicated for E. coli genotyping (Achtman and Pasteur schemes or the method proposed by Clermont) the genes coding for specific proteins (usually house-keeping genes) are amplified and differences of sequences of these genes are the base for differentiation of the strains tested. Herein we wanted to develop the method of differentiation of E. coli strains tested only on the basis of differences of sizes of amplified fragments of DNA – thus in our opinion the information what is encoded in the amplified fragment of DNA (or knowing sequences of amplified fragments) is not crucial for our method. 

Add 3. Authors compared the discriminatory power of their novel methods with other known typing techniques. The novel techniques are able to distinguish various E. coli strains on the same or better level than generally used methods, but the authors did not show the cluster analysis for these various techniques. Are dendrograms similar or different for methods presented in Table 1? These results can confirm or not the relevance of genomic regions used for novel genotyping.

The dendrograms presenting results of differentiation of 15 strains of E. coli (genomes of these strains were used for designing of primers for our methods) with Achtman and Pasteur schemes,  Clermont method, analysis of whole genomes and new MLST method are included to the manuscript and are presented as Supplementary materials – Figure 4. 

Add 4. Besides comparisons of dendrograms from various methods (see above), dendrograms presented in this manuscript should be completed with “fingerprints” (PCR profiles) used for constructions of these trees. For example, Fig. 1 and 2 should be combined and presented together. Moreover, the fingerprints of BOX-PCR have a poor quality and bands used in cluster analysis should be marked.

Unfortunately, our gels contain only 30 rows including DNA markers and combination of these gels (81 strains were analysed) would be difficult. For construction of these dendrograms we prepared tables where we included sizes of all PCR products (for each method separately). The size of PCR products was determined using software, but it can also be estimated visually. If it is possibly we would like to present results in the current form. The main point of this publication is presenting two new methods of genotyping of E. coli. The genotyping of the group of 81 E. coli strains from the hospital located in Siedlce only confirm usefulness and high differentiation potential of newly developed methods. The most important advantages of presentation of the obtained results in the current form is the fact that figures 5-9 clearly indicate differentiation power of each of pairs of primers (and the fact that positive result of amplification was obtained for each strain tested) and figure 3 highlights the fact that results obtained in PCR-RFLP method are clear and easy for interpretation. 

However we agree with the opinion on the reviewer presented in the point 4b. We have not shown the fingerprints. In the revised version of the manuscript we have shown the sizes of all 5 products obtained in MLST method as well as sizes of all DNA fragments obtained after digestion of the amplified fragment of H antigen with RsaI restriction enzyme – for each of the genotypes presented in Table 2. The results are presented in the Table 1S – Supplementary materials. Additionally we prepared dendrogram on the basis of results of all three methods – Figure 5S (included to supplementary materials). We would be grateful if you could accept this form of presentation of obtained results.

 Add 4b. While a lot of results is presented in discussion (e.g. whole paragraph L335-389), there are not showed the fingerprints of genotypes identified in this study and presented in Table 2.  Without presenting of type-fingerprints for identified genotypes the obtained data will be not comparable with following studies. In this form, the results only showed that E. colistrains/isolates are different, but it is not possible to classify these E. coli isolates. Classification (based on similarity or phylogeny) is important part of genotyping.

Explained above – the sizes of all DNA fragments amplified in MLST method as well as fragments obtained after digestion of the amplified fragment of H antigen with RsaI restriction enzyme are presented in the Table 1S – Supplementary materials. In our opinion these results could be very helpful for comparison of the results presented in our study with results obtained by other authors.

One more comment – in our previous in silico analysis we probably did an error – with Achtman scheme only 8 different genotypes were found within the group of 15 reference strains of E. coli. New analysis revealed 12 different genotypes – it has been changed in the manuscript.

Round 3

Reviewer 2 Report

Authors succesfully answered all my comments.